# Genital GVHD in Female Children and Adolescents: A Systematic Review of Case Reports and Case Series

**DOI:** 10.3390/children10091463

**Published:** 2023-08-28

**Authors:** Maria Tsimeki, Antonios Tsimpidakis, Stella Roidi, Themos Gregoriadis, Alexandra Soldatou, Vasiliki Kitra, Lina Michala

**Affiliations:** 1First Department of Obstetrics and Gynecology, School of Medicine, Alexandra Hospital, National and Kapodistrian University of Athens, 11528 Athens, Greece; maria.tsimeki@gmail.com (M.T.); s_roidi@hotmail.com (S.R.); tgregos@yahoo.com (T.G.); 2First Department of Dermatology-Venereology, School of Medicine, Andreas Syngros Hospital, National and Kapodistrian University of Athens, 16121 Athens, Greece; tsimpidakis.antonis@gmail.com; 3Second Department of Pediatrics, School of Medicine, Children’s Hospital of Athens “P. & A. Kyriakou”, National and Kapodistrian University of Athens, 11527 Athens, Greece; 4Stem Cell Transplant Unit, “Agia Sofia Children’s Hospital” Infectious Diseases Unit, Department of Pathophysiology, School of Medicine, Laikon General Hospital, National and Kapodistrian University of Athens, 11527 Athens, Greece; vkitra@hotmail.co.uk

**Keywords:** bone marrow transplantation, genital graft-versus-host disease (GVHD), genital GVHD

## Abstract

Genital graft-versus-host disease (GVHD) after hematopoietic stem cell transplantation (HSCT) is an underdiagnosed manifestation of chronic GVHD. Few articles have been published in pediatric populations, and there are no established guidelines for the management of this condition in children. This study aims to provide a systematic literature review of the published studies and cases of genital (vulvovaginal) GVHD in girls and adolescents post HSCT, with a focus on the time of diagnosis and clinical manifestations. The authors searched for English-language articles published after 1990, which included full patient details. Thirty-two cases of female patients under 20 years of age were identified. The median time of diagnosis was 381 days (IQR: 226–730 days), and 83% of patients developed Grade 3 vulvovaginal GVHD. Based on these observations, an early pediatric gynecologic examination of these patients, soon within the first year after HSCT, could be suggested for early diagnosis, treatment initiation and prevention of long-term complications.

## 1. Introduction

Hematopoietic stem cell transplantation (HSCT) is an established curative treatment for an increasing number of patients with life-threatening hematological, oncological, hereditary and immunological diseases, with more indications continuously being added. A stem cell source for transplantation can be bone marrow (BM), peripheral blood (PB), or umbilical cord blood (CB) [1]. Children who have undergone HSCT are surviving much longer than in past decades and are therefore facing long-term sequelae that can involve several organs, including the skin, oral mucosa, eyes, liver, gastrointestinal tract and lungs [2]. 

Gynecologic sequelae in this context include thrombocytopenia-associated abnormal uterine bleeding, premature ovarian insufficiency, infertility, human papillomavirus (HPV)-associated genital tract disease, secondary gynecologic malignancy and genital graft-versus-host disease (GVHD) [3]. The last is a recognized but underestimated manifestation of chronic GVHD that may affect the vulva and vagina [4]. 

Most of the published articles on genital GVHD are based on adult populations, while in pediatric patients, only a few case series have been reported. In adult studies, the reported incidence varies to up to 66%, but nonetheless, genital GVHD appears to be an under-reported condition [4,5,6]. Three case series of pediatric populations post HSCT by Allen et al. [7], Cizek et al. [8] and, more recently, Dowlut-McElroy et al. [9] reported an incidence of 1.2%, 5.9% and 5%, respectively, though it is mentioned that these percentages are probably not reliable estimates of the true incidence. In another retrospective study, Takahashi et al. [10] did not observe any cases of genital GVHD in a population of 573 children who had undergone HSCT, mentioning again the high possibility of underdiagnosis. In fact, these three papers are the only available studies that investigated the incidence of genital GVHD in pediatric populations. Underdiagnosis in this population is very likely to be higher than in adults because genital symptoms are not mentioned frequently by children, and a routine gynecological examination is not performed in these age groups.

Symptoms of genital GVHD include vulvovaginal dryness, itching, burning sensation, pain, dysuria, dyspareunia and rarely bleeding or vaginal discharge. Clinical findings are the vulvar erythema, tenderness of the vestibule, mucosal erosions or fissures, adhesions, lace-like leukokeratosis, labial resorption, clitoral hood agglutination and narrowing and shortening of the vaginal canal even to complete vaginal stenosis and hematocolpos [3,4,11]. 

The aim of this review is to investigate the appropriate time for a pediatric gynecologic evaluation of female children and adolescents post HSCT, as well as focus on the diagnosis and treatment approach of genital GVHD in this population, as there are no established guidelines.

## 2. Materials and Methods

For this systematic review, we searched the electronic database MEDLINE (Medical Literature Analysis and Retrieval System Online) on https://pubmed.ncbi.nlm.nih.gov/, (accessed on 15 January 2023) with the following MeSH terms: (graft versus host OR GVHD OR graft vs. host) AND (child* OR adolescen*) AND (genital OR vaginal or vulvovaginal OR vulvar). This search was performed in January 2023 and yielded 86 results. We searched for primary studies and case reports/case series. Moreover, the Scopus database was searched with the same terms, yielding 39 results. With a hands-on search, we added 8 more articles to the pool of screening. The record selection process was carried out according to Prisma guidelines (Figure 1). 

Data were screened by one reviewer and crosschecked by another. Each reviewer screened the articles independently, followed by a comparison of the results and discussion to reach a consensus about final inclusions. Inclusion criteria were (1) studies (or case reports or case series) with female patients, (2) studies (or case reports or case series) with children and adolescents younger than 20 years old, (3) articles published from 1990 and after, (4) articles that included full patient characteristics and details regarding the diagnosis and treatment and (5) articles written in English. We therefore excluded articles written in a non-English language, studies/cases with adult patients (>20 years old), studies/cases with males and articles written before 1990. In the case of studies on adults, which included female adolescent cases, we identified and included the relevant patients in the final table as long as they fulfilled the other inclusion criteria.

After duplicates and articles with irrelevant content based on title were excluded, 73 abstracts were assessed for possible eligibility. After their evaluation, 31 records were excluded as non-relevant. Forty-two papers were fully assessed and analyzed. Finally, 15 articles included patients who met the three inclusion criteria. Out of these, 7 articles included detailed information on patient characteristics for every subject (age, time of diagnosis, stem cell source, donor, conditioning regimen, GVHD prophylaxis, symptoms/clinical findings and genital GVHD treatment). The remaining 8 studies included some adolescents based on demographics, but complete patient characteristics were not published [10,12,13,14,15,16,17,18] and thus did not fulfill the fourth condition of the inclusion criteria (Figure 1). A total of 32 cases were included in our table. The Stratton grading system was used [11] to evaluate disease severity and was assigned by reviewers based on the clinical findings published in each case—Grade 1 (Minimal): generalized erythema and edema of vulvar structures, patchy erythema of mucosa and glandular structures of vulvar vestibule and erythema around openings of vestibular glands; Grade 2 (Moderate): Grade 1 findings plus erosions of mucosal surfaces of the vulva as well as fissures in vulvar folds and Grade 3 (Severe): Grade 2 findings plus agglutination of the clitoral hood, introital stenosis, vaginal synechiae, hematocolpos or complete vaginal closure. 

Only articles with descriptions of full patients’ characteristics were included in the table. Nonetheless, we used the National Institute of Health (ΝΙH) Quality Assessment Tool for Case Series Studies to assess and specify the risk of bias and methodological quality of the included articles (Appendix A). The CARE Checklist was also used to evaluate the quality of case reports. Statistics were analyzed using STATA SE version 11.

## 3. Results

Thirty-two young females were included, and their full characteristics are presented in Table 1—three cases by Dowlut-McElroy et al. (2022) [9], four cases by Allen et al. (2020) [7], seventeen cases by Cizek et al. (2019) [8], two cases by Michala et al. (2018) [19], one case by Childress et al. (2015) [20], one case by Choi et al. (2009) [21] and four cases by Stratton et al. (2007) [11]. The age of the patients during the HSCT ranged from 1.4 to 20 years, with a mean of 10.9 (SD: 4.7). The time of presentation for the gynecologic evaluation and diagnosis of genital GVHD ranged from 62 days to 2966 days (8.1 years), with a median of 381 days (IQR: 226–730 days). Regarding the stem cell source, 10 patients received peripheral blood stem cells (PBSC), 18 patients received bone marrow stem cells (BMSC), 1 patient received both PBSC and BMSC and for 2 patients the source was the umbilical cord. Indication for HSCT varies and includes both malignant and non-malignant immunologic/hematologic conditions (Table 1). Most patients were treated with topical steroids. Twenty-four patients (83%) developed Grade 3 genital GVHD. On the other hand, at least 10 patients had not reported any symptoms at the time of diagnosis. A total of 23 out of 31 (71.2%) patients were receiving systematic treatment for non-genital GVHD at the time of genital GVHD diagnosis. All but one patient had suffered extra-genital GVHD. The case by Childress did not report information about other GVHD sites, GVHD prophylaxis or treatment and stem cell sources.

## 4. Discussion

The reported incidence of genital GVHD in the pediatric population, based on few published studies, has been an estimated 1.2–5.9% [7,8,9]. However, the true incidence cannot be accurately estimated due to the under-reporting of symptoms and the absence of a regular gynecologic follow-up of these patients. The recommendation for a routine clinical evaluation post HSCT could be helpful in revealing the exact incidence rate. A high percentage of the patients (83%) had a Grade 3 genital GVHD at diagnosis with diffuse adhesions and architectural distortion, showing that the disease in many cases is diagnosed late, potentially due to lack of awareness. A higher grade of GVHD means more severe clinical findings, but this is not always related to symptom severity. If the vast majority of patients were diagnosed at this stage, we can assume that diagnosis is not made early enough for those clinical findings to be prevented. The absence of symptoms in some cases indicates that clinical examination is necessary, even for patients who do not complain about genital-specific symptomatology. If no diagnostic clinical findings are present or in cases of atypical manifestations, a biopsy could be helpful to assist in establishing a diagnosis. However, vulvar biopsies in children are traumatic and thus should not be routinely performed except for the cases of a severe diagnostic dilemma. Infections and dysplasia are the main differential diagnoses to be excluded, though due to the variety of possible clinical signs, a wide list of possible conditions could resemble genital GVHD [4,6]. Regarding clinical findings, Cizek et al. [8] reported that vulvar adhesions/agglutination were present at 89%, loss of vulvar architecture at 42% and vulvar skin erosions/fissures at 37% of the patients. Less often, vestibular pain to palpation (11%), atrophic vaginal mucosa (16%), abnormal vaginal discharge (5%), vulvar skin hyperpigmentation (5%) and vulvar skin dryness/scaling (5%) were reported. In order to detect the disease earlier and avoid late diagnosis, asking these patients about vulvovaginal symptoms is undoubtedly important. Genital symptoms are under-reported, especially in the female pediatric population [6]. However, symptoms are not always present, and for this reason, clinical examination should be performed by an experienced pediatric gynecologist [6]. Vaginal examination in very young children could be difficult, and sometimes sedation is needed. For that reason, it is not routinely recommended, but it could be performed in cases with clinical evidence of vaginal involvement. 

In this review, 13 out of the 32 patients developed genital GVHD within the 1st year after HSCT, whereas 3 of them developed it in less than 3 months. A total of 75% of the cases developed genital GVHD within a period of 2 years. This highlights the fact that a gynecological evaluation, performed by a gynecologist/specialist, of these patients should be performed soon after HSCT and then a regular follow-up should be suggested. Allen et al. [7] reported that the incidence of genital GVHD among those who developed GVHD in any organ system was 5%, while among all patients in the post-HSCT pediatric population, this incidence was 1.2%. In our review, we found that 27 of 28 patients had a history of acute or chronic GVHD at other sites (skin was the most common), and the majority had received immunosuppressive treatment. This suggests that children who develop GVHD at any organ site should be closely evaluated for the development of genital GvHD, with a review of genital symptoms and an age-appropriate genital examination. While peripheral blood stem cell transplantation (PBSCT) is reported to increase the risk of genital chronic GVHD, compared to bone marrow stem cell transplantation (BMSCT) [13,18], the findings of this review show that the majority of patients had undergone BMSCT. However, more cohort studies are needed to evaluate the incidence and risk factors of genital GVHD. Regarding treatment, topical steroids have mostly been used in the studied cases and included high-, medium- and low-potency agents; clobetasol, triamcinolone, betamethasone, beclamethasone, mometasone and hydrocortisone were among those. Super potent clobetasol was the most frequently used topical agent. Topical tacrolimus and topical estrogen creams or gels were also prescribed, as well as barrier creams and emollients. Topical estrogen is only used after puberty and should be administered concomitantly with systemic hormone replacement therapy if this is required. In one case of severe vaginal stenosis and hematocolpos, a vaginal dilator was used at the age of 15 [19]. Two cases required surgical lysis of adhesions [20,21], and in one of these, the use of dilators was recommended thereafter. Twenty patients used topical steroids, and another ten patients used topical estradiol or estriol. Most of the patients had some degree of improvement, and only two are clearly mentioned to have refractory disease. Apart from topical corticosteroids, topical estrogens and topical calcineurin inhibitors, other treatment options for adult patients include reduction in mechanical or chemical irritation of the genital skin by using only warm water for genital hygiene and avoidance of perfumed lotions, soaps and tight underwear. These could be also applied to the pediatric and adolescent populations. In our opinion, genital inspection by an experienced clinician, ideally within a pediatric and adolescent gynecology setting, should be planned and performed at regular intervals after stem cell transplantation, especially for those patients who have any form of GVHD manifestation. Prompt identification and treatment with topical measures should prevent the progression of the disease and non-reversible vulval and genital architectural changes. Although this review provides important information, results must be interpreted with caution and several limitations should be borne in mind. The small number of published cases, heterogeneity of the available publications and the retrospective nature of the studies are the main limitations of this study. Furthermore, there is a high possibility of publication bias. Cases with milder or no symptoms are not easily identified and not frequently published, especially as single case reports. Also, there is a possible selection bias, because eight studies were excluded as they did not provide detailed information and did not fulfill inclusion criterion 4. Large prospective studies with the long-term follow-up of children after HSCT with emphasis on genital symptoms and signs should be performed, to reach conclusions regarding the epidemiology and management of genital GVHD in these patients. Pediatric genital GVHD has been poorly investigated so far, and this review was conducted in order to raise alert and to summarize the available data. 

## 5. Conclusions

In conclusion, early identification of genital GVHD and subsequent initiation of treatment may prevent severe disease. Left untreated, genital GVHD will impact the girls'’future reproductive health and will seriously affect their quality of life. Given the lack of established guidelines regarding its management and treatment, further prospective studies are needed in this direction.

## Figures and Tables

**Figure 1 children-10-01463-f001:**
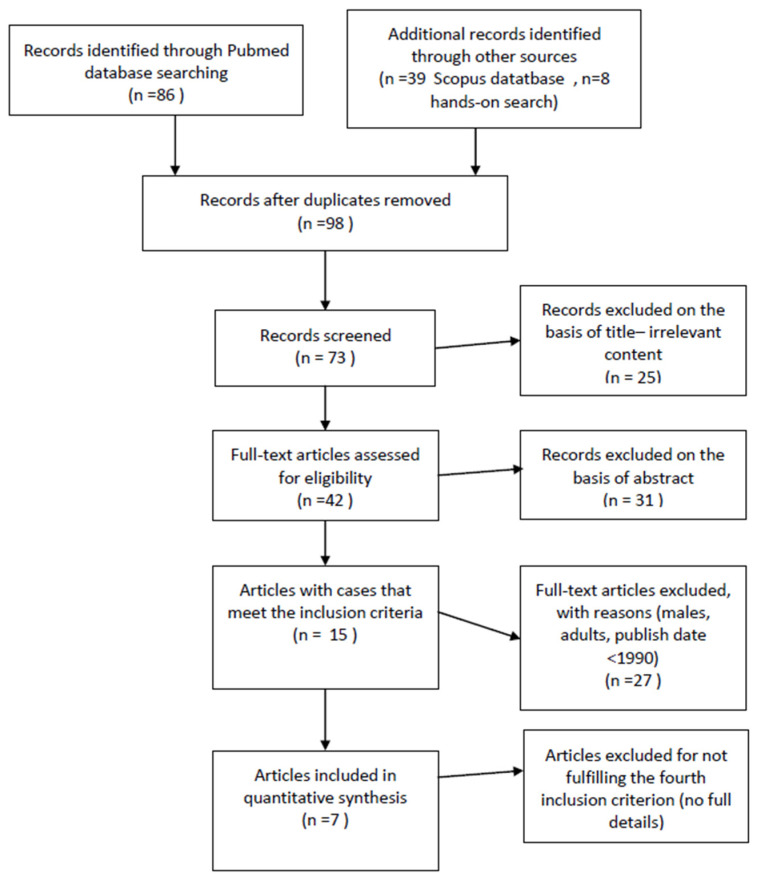
Selection PRISMA flow chart.

**Table 1 children-10-01463-t001:** Results. Full patients’ characteristics.

Study	Patient	Diagnosis	Age at HSCT	Stem Cell Source	Related/Unrelated Donor; Match	Conditioning Regimen	GVHD Prophylaxis	Acute GVHD Site	Chronic GVHD Severity/Site	Non-Genital GVHD Treatment at Diagnosis of Vulvovaginal GVHD	Symptoms	Clinical Findings	Treatment of Genital GVHD	Long-Term Follow-Up/Response	Time of vvGVHD Diagnosis after HSCT	Genital GVHD Grading
Dowlut-McElroy et al., 2022 [9]	1	Infantile osteopetrosis	1.4	BM	N/A	N/A	Sirol	None	Lungs	Sirol	Unknown	Unknown	Premarin cream	Unknown	N/A	Unknown
	2	Acute myelogenous leukemia	12	PBSC	N/A	N/A	Sirol	Skin/Vulva	Skin	Sirol	None	Erythema	Cream Betamethasone, ointment pimecrolimus	CR	62	3
	3	Acute myelogenous leukemia	4	UC	N/A	N/A	Sirol	Skin	Skin	Sirol	Dysuria	Erythema, mild adhesions	None	CR	1565	1
Cicek et al., 2019 [8]	4	Acute myelogenous leukemia	2.4	BM	Unrelated	ATG/BU/CY	CSA/MTX	Skin	Skin, eyes	Pred	Vulvar pruritus, dysuria	Not specified for each case. Clinical findings of the whole Study: vulvar adhesions/ agglutination (89%), loss of vulvar architecture (42%), vulvar skin erosions/ fissures (37%), vestibular pain to palpation (11%), vaginal atrophy (16%), abnormal vaginal discharge (5%), vulvar skin hyperpigmentation, (5%) and vulvar skin dryness/scaling (5%)	Beclomethasone 0.05%, tacrolimus topical, estradiol cream, betamethasone	PR with relapse	287	3
	5	Severe combined immunodeficiency	3.8	BM	Unrelated	Cam/Flu/Mel	CSA/Pred	None	MS	None	No symptoms		Betamethasone cream	Unknown	2966	3
	6	Chronic myelogenous leukemia	8.7	BM	Unrelated	ATG/CY/TBI	CSA/Pred	None	Intestine, liver, oral mucosa	Pred	Vulvar pain/Pruritus, dysuria		Betamethasone dipropionate ointment 0.1%	PR	452	3
	7	Dyskeratosis congenita	9.3	BM	Unrelated	Cam/Flu/Mel	CSA/MMF	Skin	Oral mucosa, skin	Rituximab, MP, MMF	Vulvar pain, dysuria		Triamcinolone 0.1% ointment, hydrocortisone 1% ointment, estradiol cream	PR	71	2
	8	Fanconi anemia	10.2	UC	Unrelated	ATG/CY/Flu/TBI	CSA/Pred	None	Skin, oral mucosa, MS	Tac	Abnormal vaginal discharge		Triamcinolone 0.1% ointment, clobetasol 0.05% ointment, estradiol cream	PR with relapse	1811	3
	9	Dyskeratosis congenita	11.9	BM	Unrelated	Cam/Flu/Mel	CSA/Pred	Skin	Skin, oral mucosa	Hydrocortisone, Pred	Vulvar pain/pruritis, abnormal vaginal discharge, dysuria, urge urinary incontinence		Clobetasol 0.05% cream, triamcinolone 0.1% ointment, estradiol cream, surgical lysis of adhesions	PR	690	3
	10	Beta thalassemia	12.2	BM	Unrelated	BU/Flu/Thio	Aba/Pred	None	Skin, lung	Infliximab, Ruxolitinib, tac, pred, MP, phototherapy	Vulvar pain/pruritus, abnormal vaginal discharge, dysuria		Clotrimazole, betamethasone 0.5% ointment	CR	600	3
	11	Acute lymphoblastic leukemia	12.8	PBSC	Unrelated	CY/TBI	CSA/MTX	Skin	Oral mucosa	Bud po	No symptoms		Estradiol cream	CR	226	3
	12	Beta thalassemia	13.2	BM	Related	BU/Flu/Thio	CSA/Mar/Pred	None	Oral mucosa, lung	MMF, pred, bud po	Vulvar pain, dysuria		Triamcinolone 0.1% ointment, betamethasone 0.1% ointment	CR	313	3
	13	Acute myelogenous leukemia	13.3	BM	Unrelated	Flu/Mel	CSA/MTX	Skin	Intestine, skin	Infliximab, pred, CSA, bud po	Vulvovaginal pain		Betamethasone 0.1% ointment, betamethasone 0.05% ointment	NR	132	1
	14	Acute lymphoblastic leukemia	13.8	BM	Unrelated	CY/TBI	Aba/CSA/Pred	Skin	Skin	Ruxolitinib, bud po	No symptoms		None	n/a	593	3
	15	Fanconi anemia	14.6	BM	Unrelated	ATG/BU/CY/Flu	CSA/Pred	Skin	Oral mucosa	None	No symptoms		Betamethasone 0.05% ointment, estradiol cream	PR	677	3
	16	Acute lymphoblastic leukemia	12.9	BM	Related	ATG/CY/TBI	MTX/Sirol/TAC	None	Skin	None	No symptoms		Clobetasol 0.05% ointment, estradiol cream	PR	381	3
	17	Fanconi anemia	10.6	PBSC	Unrelated	ATG/BU/CY/Flu	CSA/TCD	None	None	None	No symptoms		Clobetasol 0.05% ointment	CR	2453	3
	18	Fanconi anemia	6.5	BM	Related	ATG/CY/Flu	ATG/CSA/MP	Gut	Lung	None	Vulvar pain/pruritus, dysuria		Clobetasol 0.05% ointment	PR	2286	3
	19	Acute myelogenous leukemia	10.2	BM	Unrelated	BU/CY/ATG	CSA/MTX	Skin	Oral mucosa	Oral bud	Unknown		None (declined)	Unknown	290	3
	20	Acute lymphoblastic leukemia	5.0	PBSC	Related; fully matched	TBI-based	Unknown	None	Skin, joints, eyes, lungs, oral mucosa	MMF, pred	No symptoms		Triamcinolone 0.1% ointment	PR	77	3
Michala et al., 2018 [19]	21	Fanconi anemia	10.5	BM	Unrelated	BU/CY, no TBI	ATG/CSA/MTX	None	Mild cGVHD skin/mucosa, conjunctivitis, stomatitis,	None (symptomatically)	No symptoms,	Hematocolpos, vulvar atrophy, clitoral phimosis	Oral estradiol, hydrocortisone foam, estriol gel, dilator	CR	1278	3
	22	Myelodysplastic syndrome	13.0	BM	Unrelated; fully matched	Be/Mel/Flu	ATG/CSA/MTX	None	Skin	Oral steroids	No symptoms,	Hematocolpos, vaginal stenosis,	Oral estradiol, hydrocortisone foam, estriol gel, Vaginal dilator	CR	456	3
Allen et al., 2020 [7]	23	Griscelli Syndrome and Hemophagocytic Lymphohistiocytosis	2.0	BM	Unrelated; matched	N/A	CSA, MP, Tac	None	Intestine, skin	Tac, etanercept, bud, MP, topical tac and triamcinolone	Vulvar pruritus,	Labial scar tissue	Barrier cream	PR	2207	3
	24	Acute myelogenous leukemia/myelodysplastic syndrome	7	PBSC	Haploidentical matched	N/A	Unknown	Skin	Skin	Steroids	Dysurea, vulvar pain, rash,	White plaque on labia	Topical triamcinolone and tacrolimus, transdermal estradiol	PR	88	1
	25	Acute myelogenous leukemia	17	BM, PBSC	1. Matched unrelated; 2. Haploidentical	N/A	Tac, steroids after BMT; tac, mycophenolic, CY	Gut	Intestine, skin, liver,	MP, Tocilizumab, rituximab, etanercept, basiliximab and topical clobetasol	Vaginal pain, labial hypopigmentation, mons and labia tightness	Labial hypopigmentation, tenderness to palpation	Topical clobetasol and mometasone	PR	284	2
	26	Acute myelogenous leukemia	17	BM	Unrelated; matched	N/A	CSA, steroids	Gut	Intestine, skin, mucosa, lung	MP, pred, etanercept, ruxolitiab, oral bud, imatinib, topical Tac, unspecified steroids (IV, topical)	Vulvar pain, sensitivity, dryness	Hypoestrogenic vulva, labial adhesions/agglutination	Hormone replacement, topical estrogen and zinc oxide barrier creams	NR	512	3
Choi et al., 2009 [21]	27	Acute myelogenous leukemia	14	PBSC	Matched unrelated	N/A	N/A	Skin, gut	None	Pred, CSA	Not reported	Labial fusion, adhesions, vulvar skin erosions	Topical steroid,	PR	730	3
Stratton et al., 2007 [11]	28	Hodgkin’s lymphoma	16	PBSC	Matched related	Flu/CY	N/A	Not specified, though all patients in the study had some manifestation of extra-genital acute or chronic GVHD		None	Yes. Not specified	Not specified for each case. Clinical findings of the whole study: vulvar erythema (93.1%), tender gland openings (89.7%), vulvar fissures or erosions (51.2%), clitoral agglutination (10.3%), vaginal synechiae (17.2%), vaginal fascitis (3.4%)	Not specified; 22 out of 29 pts were treated with 0.5%clobetasol propionate; the others were prescribed 0.025% fluocinolone acetonide or 0.01% tacrolimus	Not specified, but mentioned that all patients had some response	312	3
	29	Acute lymphoblastic leukemia	9	PBSC	Matched related	Flu/CY	N/A			Mycophenolate, pred, extracorporeal photophoresis	Yes. Not specified				374	.
	30	Myelodysplastic syndrome/Ewing Sarcoma	20	PBSC	Unrelated; cord blood	Bu/Cy/ATG	N/A			Pred, CSA	Yes. Not specified				224	.
	31	Ewing Sarcoma	19	PBSC	Matched related	Flu/CY	N/A			None	Yes. Not specified				80	3
Childress et al., 2015 * [20]	32	Aplastic anemia	12	N/A	N/A	N/A	N/A	N/A	N/A	N/A	Discharge	Erythema and erosions, discharge, adhesions, ematocolpos	Topical clobetasol, surgical, topical estrogen	NR, then PR	365	3

Tables Abbreviations: BM, bone marrow; UC, umbilical cord; PBSC, peripheral blood stem cells; ATG, antithymocyte globulin; BU, busulfan; CY, cyclophosphamide; Cam, Campath; Flu, fludarabine; Mel, melphalan; TBI, total body irradiation; Thio, thiotepa; CSA, cyclosporine; MTX: Methotrexate; Pred, prednisone; MMF, mycophenolate mofetil; Bud po, budesonide oral swish and spit; Aba, abatacept; Mar, maraviroc; Sirol, sirolimus; Tac, tacrolimus; TCD, T cell depletion; MP, methylprednisolone; CR, complete response, PR partial response, NR: no response. * Severity grading per NIH consensus criteria, The case by Childress did not report information about other GVHD sites, GVHD prophylaxis or treatment and stem cell source.

## Data Availability

Not applicable.

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
