# Peer review of "Genital GVHD in Female Children and Adolescents: A Systematic Review of Case Reports and Case Series"

_children, 2023, doi:10.3390/children10091463_

Round 1

Reviewer 1 Report

The article needs a large amount of work in my opinion, to rise to the journal’s standards, but I believe the authors can manage in succeeding.

First, the article is sloppily written. You have more extra spaces. Please, remove extra spaces between words, space before or/and after paragraph. Please use justified alignment.

Second, please format the article according MDPI requirements.

Abstract:

line 24: please write patients under

The Introduction builds a logical context for the article, the purpose of which is clear and well-articulated. But I suggest doing some correction:

- please remove paragraph 0. How to use this template

- please rephase female children

In Materials and Methods, you write to the inclusion criteria that studies (or case reports 88 or case series) with female patients, but in exclusion criteria you write the same studies/cases with adult patients (>20 years old). Please clarify.

Please, write very clearly this part of article. It isn't understandable how the articles were excluded, and their number isn't very clear. Please describe the statistical analysis you used.

Results

Please remove line 136: 3.1. subsection

Please remove line 137

Please specify the risk of bias.

The discussion segment is treated briefly and superficially. It should be extended with a more complete analyze of literature and critical appraisal.

The references are not written according MDPI requirements.

The written English must be corrected by a native and/or professional writer/editor who is experienced in producing standardized research papers. 

Author Response

Dear reviewer thank you very much for the time allocated in reviewing our paper and the comments provided, which were really helpful in improving our manuscript.

A point by point response with text in bold:

The article needs a large amount of work in my opinion, to rise to the journal’s standards, but I believe the authors can manage in succeeding.

First, the article is sloppily written. You have more extra spaces. Please, remove extra spaces between words, space before or/and after paragraph. Please use justified alignment. Extra spaces have been deleted and paragraphs fixed

Second, please format the article according MDPI requirements. format has been corrected accordingly

Abstract:

line 24: please write patients under

The Introduction builds a logical context for the article, the purpose of which is clear and well-articulated. But I suggest doing some correction:

- please remove paragraph 0. How to use this template.  has been removed

- please rephase female children rephrased

In Materials and Methods, you write to the inclusion criteria that studies (or case reports 88 or case series) with female patients, but in exclusion criteria you write the same studies/cases with adult patients (>20 years old). Please clarify.

In the case of  studies on adults that included female adolescent cases, we identified and included the reelevant patients in the final table as long as they fulfilled the other inclusion criteria.  

Please, write very clearly this part of article. It isn't understandable how the articles were excluded, and their number isn't very clear. Please describe the statistical analysis you used.

Thank you, the methods section has been updated. The flow chart has been revised as there were two mistakes: In the first exclusion block the correct number is 25 instead of 19 (Records excluded on the basis of title– irrelevant content (n = 25)). Moreover the second exclusion block was revised to: “Records excluded on the basis of abstract” instead of title and abstract. The text has been also revised to be understandable.

Statistics were analyzed using STATA SE version 11.

Results

Please remove line 136: 3.1. subsection done

Please remove line 137 done

Please specify the risk of bias.

Thank you very much for the comment. We have included the following paragraph in the methods section “Only articles with description of full patients’ characteristics were included in the table. Nonetheless we used the National Insitute of Health Quality Assessment Tool for Case Series Studies to assess and specify the risk of bias and methodological quality of the included articles (Supplementary table 1). CARE Checklist was also used to evaluate quality of case reports. “ 

We have also uploaded a supplementary table.

The discussion segment is treated briefly and superficially. It should be extended with a more complete analyze of literature and critical appraisal.

Discussion has been revised

The references are not written according MDPI requirements. 

Have been updated thank you (DOI was missing)

Comments on the Quality of English Language

The written English must be corrected by a native and/or professional writer/editor who is experienced in producing standardized research papers. 

Thank you, English have been corrected by professional wrtiter. 

Reviewer 2 Report

Since the literature has scarce data, this study tries to identify the mechanisms that will underlie the best experimental and clinical practices.

Some suggestions could improve the quality of the article:

- Lines 160-162, please be more specific - to which patients this biopsy is addressed? Vulvar biopsies in children are not routinely performed as they are traumatic.

- The vaginal examination of these patients requires sedation.

- To discuss the particularities of the symptomatology and, respectively, of the treatment in children versus adolescents.

- The conclusions are unsystematized and must be summarized.

- The main ideas of this study related to clinical practice should be pointed out at the end of the article as a take-home message.

Kind regards

Author Response

Dear reviewer thank you very much for your comments

point by point response (text in bold)

Some suggestions could improve the quality of the article:

- Lines 160-162, please be more specific - to which patients this biopsy is addressed? Vulvar biopsies in children are not routinely performed as they are traumatic. The following sentence has been added: "However vulvar biopsies in children are traumatic and thus should be not routinely performed except for the cases of severe diagnostic dilemma"

- The vaginal examination of these patients requires sedation. 

it is true that vaginal examination in very young children is not routinely recommeded and kept only for cases were there is strong clinical evidence of vaginal involvement . The text has been revised accordingly. 

- To discuss the particularities of the symptomatology and, respectively, of the treatment in children versus adolescents.

thank you very interesting comment, however there are so limited published cases that we think that we cannot draw conclusions on this. 

- The conclusions are unsystematized and must be summarized.

conclusion has been revised

- The main ideas of this study related to clinical practice should be pointed out at the end of the article as a take-home message.

Thank you, Conclusion has been shortened and revised in order to focus on the take home message. The rest of the text has been moved to the discussion section.

Reviewer 3 Report

Esteemed authors and editorial team, this article provides valuable insight into a pathology quite out of the ordinary and raises awareness of both parents and gynecologists regarding this potential clinical encounter. The article is well designed and impeccably written.

A minor comment regards a spelling error at line 199 - correct delators to dilators.

1. The target of the study is to raise awareness and make an up-to-date review regarding a rare pathology, genital graft versus host disease (GVHD) after hematopoietic stem cell transplantation in the pediatric population.

This is a welcome subject since such cases are rare, underrecognized and underreported. All gynecologists should be warned regarding the potential occurrence of such cases. More so in the pediatric/young female population which are not that often in the eye of the gynecologist and which may require a special anamnesis/insistence to highlight specific symptoms and offer relief as well as prevention of progression of disease.

2. As I had already stated, in my original comments and above, it is essential to emphasize the specials of this condition in the population taken into discussion. 

3. Literature regarding genital graft versus host disease is scarce. Having “experience” with such cases is rare. And overlooking symptoms leads to debilitating sequalae. Management of pediatric cases comes with psychologic and ethical implications which are fully addressed by the authors.

4. In the context of under-recognition and/or underreporting, there is perhaps an important educational gap of professionals. In my opinion, methodology is adequate for the study design.

5. Conclusions are quite stuffy but present a good synopsis of the ideas vehicle by the research.

6. References are on point and uptodate. No relevant recent literature seems to be missing.

7. Both the PRISMA flow-chart and table look fine.

Author Response

Dear reviewer thank you very much for the comments

point-by-point response (text in bold)

A minor comment regA minor comment regards a spelling error at line 199 - correct delators to dilators. Has been corrected thank you

. The target of the study is to raise awareness and make an up-to-date review regarding a rare pathology, genital graft versus host disease (GVHD) after hematopoietic stem cell transplantation in the pediatric population.

This is a welcome subject since such cases are rare, underrecognized and underreported. All gynecologists should be warned regarding the potential occurrence of such cases. More so in the pediatric/young female population which are not that often in the eye of the gynecologist and which may require a special anamnesis/insistence to highlight specific symptoms and offer relief as well as prevention of progression of disease.

  1. As I had already stated, in my original comments and above, it is essential to emphasize the specials of this condition in the population taken into discussion.
  2. Literature regarding genital graft versus host disease is scarce. Having “experience” with such cases is rare. And overlooking symptoms leads to debilitating sequalae. Management of pediatric cases comes with psychologic and ethical implications which are fully addressed by the authors.
  3. In the context of under-recognition and/or underreporting, there is perhaps an important educational gap of professionals. In my opinion, methodology is adequate for the study design.
  4. Conclusions are quite stuffy but present a good synopsis of the ideas vehicle by the research. We made a revision in conclusion and discussion section
  5. References are on point and uptodate. No relevant recent literature seems to be missing.
  6. Both the PRISMA flow-chart and table look fine.

Thank you very much for the time allocated in reviewing our manuscript

kind regards

Round 2

Reviewer 1 Report

 You have more extra spaces. Please, remove extra spaces between words, space before or/and after paragraph. 

The references are not written according MDPI requirements. 

Reviewer 2 Report

Dear authors,

There is scarce data in the literature describing the effects of genital GVHD in female children and adolescents. The article in this form does provide adequate details of the methodology, evaluation, findings, and investigations, which are well outlined in the results and conclusions sections.

Kind regards